# Imitating Radiological Scrolling: A Global-Local Attention Model for 3D Chest CT Volumes Multi-Label Anomaly Classification

**Theo Di Piazza**[1,2]               THEO.DIPIAZZA@CREATIS.INSA-LYON.FR
**Carole Lazarus**[3]
**Olivier Nempont**[3]
**Loic Boussel**[1,2]

[1] *UCBL1, INSA Lyon, CNRS, INSERM, CREATIS UMR 5220, U1294, Villeurbanne, France*

[2] *Department of Radiology, Croix-Rousse Hospital, Hospices Civils de Lyon, Lyon, France*

[3] *Philips Clinical Informatics, Innovation Paris, France*

**Editors:** Accepted for publication at MIDL 2025

## Abstract

The rapid increase in the number of Computed Tomography (CT) scan examinations has created an urgent need for automated tools, such as organ segmentation, anomaly classification, and report generation, to assist radiologists with their growing workload. Multi-label classification of Three-Dimensional (3D) CT scans is a challenging task due to the volumetric nature of the data and the variety of anomalies to be detected. Existing deep learning methods based on Convolutional Neural Networks (CNNs) struggle to capture long-range dependencies effectively, while Vision Transformers require extensive pre-training, posing challenges for practical use. Additionally, these existing methods do not explicitly model the radiologist's navigational behavior while scrolling through CT scan slices, which requires both global context understanding and local detail awareness. In this study, we present CT-Scroll, a novel global-local attention model specifically designed to emulate the scrolling behavior of radiologists during the analysis of 3D CT scans. Our approach is evaluated on two public datasets, demonstrating its efficacy through comprehensive experiments and an ablation study that highlights the contribution of each model component.

**Keywords:** Multi-label classification, Computed-Tomography, Attention Mechanism.

## 1. Introduction

Computed Tomography (CT) provides detailed imaging of the human body, enabling radiologists to thoroughly examine various anatomical regions, identify abnormalities, and guide patient care from initial diagnosis to follow-up (Mazonakis and Damilakis, 2016). However, the growing number of CT scans (Broder and Warshauer, 2006) and the associated workload for radiologists have created a pressing need for automated methods to assist in analyzing these volumes (Chen et al., 2022). In medical imaging, and particularly in CT scans, substantial progress has been made in leveraging deep learning techniques to support radiologists in tasks such as segmentation (Gu et al., 2022), image restoration (Yuan et al., 2023), classification (Draelos et al., 2021), and more recently, report generation (Hamamci et al., 2024b). As illustrated in Figure 1, multi-label anomaly classification from Three-Dimensional (3D) CT volumes remains a challenging task due to the significant variability in the anomalies that need to be detected.

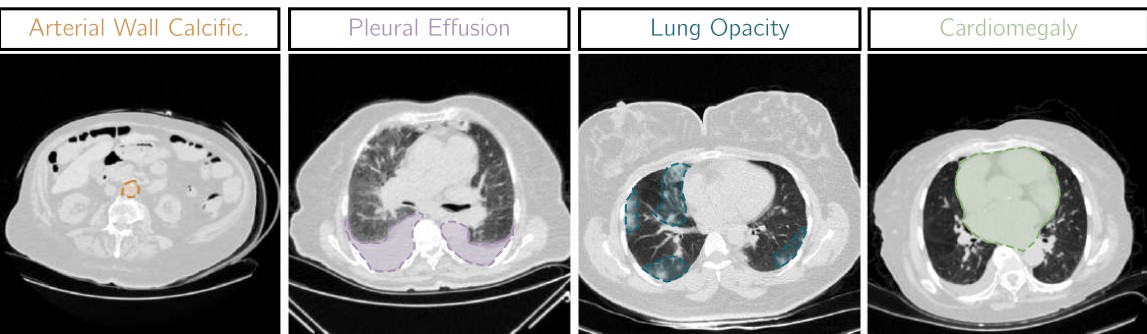

Figure 1: Examples of 4 axial CT scan slices with anomalies of varying sizes from the CT-RATE dataset.

To process volumetric data and extract meaningful visual features, early approaches predominantly relied on 3D convolutional neural networks (CNNs) to capture spatial dependencies within volumetric data effectively (Singh et al., 2020). Alternatively, some studies adopted conventional 2D architectures by treating a volume as a sequence of slices and subsequently fusing the extracted features (Draelos et al., 2021). CNNs excel at capturing local spatial features, and their hierarchical structure facilitates the progressive learning of features, from low-level patterns to high-level semantic representations. More recently, attention mechanisms (Vaswani et al., 2023), initially introduced in Natural Language Processing, have demonstrated exceptional performance across diverse text-related tasks (Touvron et al., 2023). This paradigm has been adapted to visual data, including 2D and 3D imaging, by representing images as sequences of 1D tokens derived from flattened 2D or 3D patches. In particular, Vision Transformers (ViTs) (Dosovitskiy et al., 2021) leverage attention mechanisms to model global context by enabling interactions across different regions of an image. This capability is particularly advantageous for applications requiring a comprehensive understanding of global contexts, making ViTs a promising alternative for complex medical imaging tasks. However, the local receptive fields of CNNs limit their ability to capture global contextual information across large 3D volumes, while ViTs can be computationally expensive when applied to high-dimensional volumetric data and often require large-scale pre-training on extensive datasets to achieve competitive performance (?). When radiologists analyze a CT scan, they typically navigate through axial slices to have a global understanding of the volume before focusing on specific anatomical regions of interest (Goergen et al., 2013). If an area appears abnormal, the radiologist often revisits the same slices repeatedly, carefully examining the local context to confirm the diagnosis. Inspired by this diagnostic approach and leveraging the strengths of alternating attention (Warner et al., 2024), originally introduced in NLP, we present a novel alternating global-local attention module, termed the *Scrolling Block* (SB), illustrated in Figure 2. This module integrates both global and local information through a Sliding Window Attention (SWA) (Child et al., 2019; Beltagy et al., 2020) mechanism specifically designed for 3D CT volumes. Our contributions are summarized as follows:

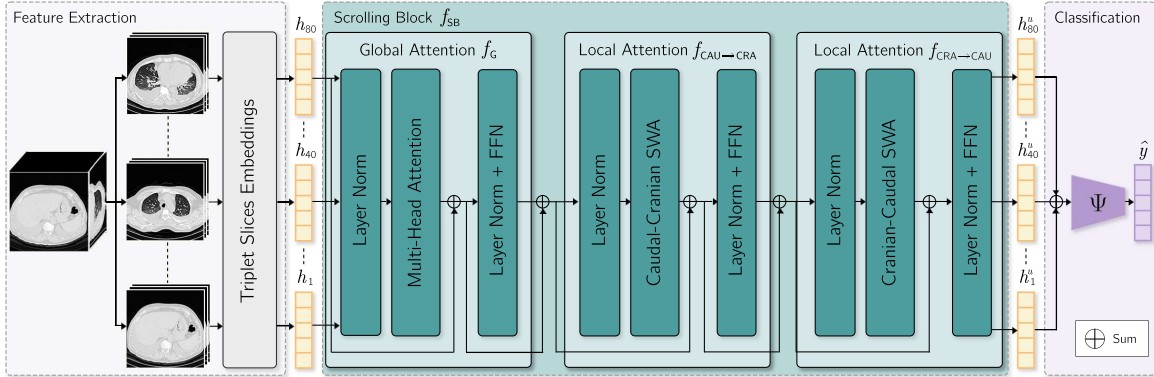

Figure 2: The CT-Scroll architecture consists of three main components. (1) Axial slices of the volume are grouped into triplets and processed by a ResNet followed by a GAP layer, producing a vector representation per triplet. (2) The Scrolling Block then refines these embedded visual tokens using both global and local attention mechanisms. (3) Finally, the aggregated features are fed into a classification head to predict anomalies.

- We propose a global-local attention model designed to imitate radiology navigation in 3D CT scans, enhancing multi-label anomaly classification while being achievable with limited computational resources (single GPU, < 24-hour training time).
- We demonstrate the robustness and generalizability of our approach through comprehensive cross-dataset evaluation on two public 3D chest CT datasets.
- We conduct a comprehensive ablation study analyzing feature reduction and aggregation modules, attention field sizes on model performance and computational efficiency. Our analysis demonstrates how varying the spatial extent of attention mechanisms in our global-local module affects the model's ability to capture multi-scale anomalies in 3D CT volumes.

## 2. Related Work

### 2.1. 3D Visual Encoder for Medical Imaging

In the domain of 3D feature extraction, significant efforts have been made across various application areas such as remote sensing, robotic manipulation, and autonomous driving (Sarker et al., 2024). In medical imaging, particularly with 3D CT scans, conventional 3D convolutional neural networks have been widely employed for segmentation (Ilesanmi et al., 2024) and classification (Ho et al., 2021) tasks. However, CNNs fundamentally lack the ability to model long-range dependencies, limiting their capacity for global context understanding (Ma et al., 2024). More recently, the adaptation of Vision Transformers (Chen et al., 2021) and Swin Transformers (Yang et al., 2023) to 3D volumes has enabled the interaction of visual tokens corresponding to different patches of the volume via self-attention mechanisms. Positional embeddings are used to preserve spatial information, facilitating better understanding of the volume structure. More recently, CT-ViT (?) was introduced

as a 3D-CT Vision Transformer used to generate 3D CT volumes from free-form medical text prompts. CT-ViT learns compact latent representations of 3D volumes by leveraging self-attention and causal attention mechanisms to address the challenges posed by CT scans with varying cranio-caudal coverage. However, Vision Transformers require extensive pre-training on large-scale datasets and a high number of parameters to be effective, which limits their practical applicability. To face this issue, CT-Net (Draelos et al., 2021) proposes grouping consecutive slices of a CT volume into triplets, which are then passed through a ResNet (He et al., 2015) followed by a small 3D CNN to extract a compact vector representation, subsequently fed into a classification head. This approach showcases robust performance in multi-label classification by effectively capturing fine-grained details, while remaining computationally efficient. However, its reliance on a 3D CNN for feature map reduction limits its ability to model long-range dependencies which can be crucial to capture broader anatomical structures. In this work, we introduce a global-local attention module that enables the modeling of both short-range and long-range dependencies accross slices along the z-axis.

## 2.2. Global and Local Attention

**Global Attention.** In both Natural Language Processing (NLP) and computer vision, Transformer-based models leverage global attention (Luong et al., 2015), where each embedded token interacts with all other tokens through the self-attention mechanism. This allows for comprehensive contextualization, capturing long-range dependencies and integrating global semantic information into the token representations (Devlin et al., 2019).

**Local Attention.** Despite its effectiveness, global attention suffers from quadratic complexity with respect to sequence length, making it computationally expensive for long sequences in NLP (Beltagy et al., 2020). To address this, local attention mechanisms such as windowed attention were introduced, restricting each token's receptive field to a local neighborhood, thereby improving efficiency while preserving essential contextual information. In computer vision, local attention has been successfully adapted in models like Swin Transformer (Liu et al., 2021), where image patches interact within localized windows. This hierarchical approach enables efficient processing of high-resolution images and enhances the model's ability to handle objects with varying scales.

**Alternating Attention.** Recent advancements in large language models (LLMs) (Touvron et al., 2023) have demonstrated the benefits of alternating global and local attention to improve efficiency and contextual modeling. For instance, ModernBERT (Warner et al., 2024) integrates architectural innovations inspired by recent LLMs (Team, 2024), alternating between global and local attention layers to balance long-range context aggregation with fine-grained local dependencies.

## 3. Dataset

**CT-RATE dataset.** We leverage the publicly available CT-RATE dataset (Hamamci et al., 2024a) to train and evaluate our proposed method. This dataset comprises 3D non-contrast chest CT scans, annotated with 18 anomalies extracted from radiology reports using a RadBERT classifier (Yan et al., 2022). The dataset is partitioned as follows: 17,799

unique patients for the train set, 1,314 unique patients for the validation set and 1,314 unique patients for the test set.

**Rad-ChestCT dataset.** To assess cross-dataset generalization, we extend our evaluation to the Rad-ChestCT test dataset (Draelos et al., 2021), which consists of 1,344 3D non-contrast chest CT scans annotated with 83 anomalies extracted using a SARLE labeler from radiology reports. Among these anomalies, we evaluate our method on the 16 anomalies shared with the CT-RATE dataset.

**Preprocessing.** For all experiments and for both datasets, all CT volumes are preprocessed to ensure uniformity and consistent input characteristics across datasets, enabling robust training and evaluation. Each volume is either center-cropped or padded to achieve a resolution of $240 \times 480 \times 480$ with an in-plane resolution of 0.75 mm and 1.5 mm in the z-axis. Hounsfield Unit values are clipped to the range $[-1000, +200]$, before normalization to $[-1, 1]$.

## 4. Method

When a radiologist navigates along the longitudinal axis of a CT volume (Patel and De Jesus, 2024), they scroll through axial slices to detect anomalies. Initially, they perform a global assessment to develop a comprehensive understanding of the volume before revisiting specific slices that may contain abnormalities. Upon identifying a potential anomaly, radiologists frequently scroll back and forth across adjacent slices to incorporate local contextual information, refining their assessment by leveraging both global structure and local details. As illustrated by Figure 2, we propose a method that extracts vector representations from triplets of slices and models their interactions using global and local attention blocks. These attention mechanisms are designed to imitate the scrolling behavior of radiologists, capturing global and local contextual relationships across CT axial slices. The extracted features are then fused to predict the presence of anomalies effectively. The model is trained for multi-label classification using a binary cross-entropy loss function (Goodfellow et al., 2016).

### 4.1. Triplet Slices Embedding

Similar to CT-Net (Draelos et al., 2021), the slices of the initial volume $x \in \mathbb{R}^{240 \times 480 \times 480}$ are grouped in triplets, where each triplet consists of three consecutive slices. This results in a 4D tensor with dimensions $(80 \times 3 \times 480 \times 480)$. For each triplet $x_i^t \in \mathbb{R}^{3 \times 480 \times 480}$ $(i \in \{1, \ldots, 80\})$, a feature map is extracted using a ResNet (He et al., 2015) pre-trained on ImageNet (Russakovsky et al., 2015), noted $f_{\text{ResNet}}$, and passed through a Global Average Pooling (GAP) layer $f_{\text{GAP}}$ to obtain a vector representation for the triplet, noted $h_i \in \mathbb{R}^{512}$ $(i \in \{1, \ldots, 80\})$, such that:

$$h_i = (f_{\text{GAP}} \circ f_{\text{ResNet}})(x_i^t), \quad \forall \, i \in \{1, \ldots, 80\}. \tag{1}$$

We employ Global Average Pooling instead of a linear projection or a 3D reducing convolutional layer to significantly reduce the total number of trainable parameters while preserving the local information encoded in the feature maps (Li et al., 2023).

Figure 3: Causal and Sliding Window Attention Masks. A mask of shape $(n, n)$ preventing attention to certain positions. 1 indicates that the corresponding position is allowed to attend, 0 otherwise. Example with $n = 5$ and $q = 3$.

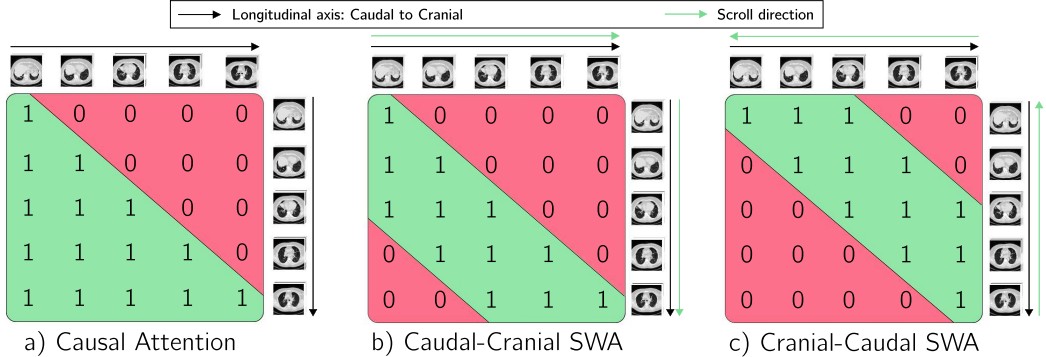

a) Causal Attention   b) Caudal-Cranial SWA   c) Cranial-Caudal SWA

## 4.2. Scrolling Block

These vector representations $h = \{h_i\}_{i=1}^{80}$, considered as visual tokens associated with the triplet slices, are then fed into a *Scrolling Block* (SB), denoted as $f_{\text{SB}}$. A Scrolling Block consists of three Transformer encoders (Vaswani et al., 2023). The first encoder, denoted as $f_{\text{G}}$, employs global self-attention, enabling each token to aggregate information from the entire volume to have a global understanding of the 3D structure by capturing long-range dependencies along the z-axis. The second and third encoders, denoted as $f_{\text{CAU}\to\text{CRA}}$ and $f_{\text{CRA}\to\text{CAU}}$, use Caudal-Cranial and Cranial-Caudal Sliding Window Attention, respectively, emulating the radiologist's scrolling behavior along the longitudinal axis to focus on local contextual information. For each triplet slice, the corresponding visual token can only interact with visual tokens associated with $q \in \mathbb{N}^+$ slices above it (for Caudal-Cranial modeling, Figure 3.b) or below it (for Cranial-Caudal modeling, Figure 3.c) along the longitudinal axis to capture short-range dependencies, as illustrated by Figure 3. Our method leverages both global attention, capturing long-range dependencies across slices, and local attention, refining contextual representations within localized regions. This design effectively models both short- and long-range interactions along the cranial-caudal axis, mirroring the way radiologists navigate through CT scans for clinical assessment. Each Transformer encoder is followed by a residual connection (He et al., 2015), a normalization layer (Ba et al., 2016), and a FeedForward Network leveraging GeGLU, a Gated Linear Unit (GLU)-based activation function that has demonstrated consistent empirical improvements over standard activation functions (Shazeer, 2020). This Scrolling Block module generates updated visual tokens with a dimension of 512, denoted as $\{h_i^u\}_{i=1}^{80}$, such that:

$$h^u = \{h_1^u, \ldots, h_{80}^u\} = f_{\text{SB}}(h) = (f_{\text{CRA}\to\text{CAU}} \circ f_{\text{CAU}\to\text{CRA}} \circ f_{\text{G}})(h). \tag{2}$$

**Aggregation.** The resulting vector representations are aggregated through summation and passed to a classification head, implemented as a lightweight Multilayer Perceptron,

denoted as $\Psi$, which predicts a logit vector $\hat{y} \in \mathbb{R}^{18}$, such as:

$$\hat{y} = \Psi \left( \sum_{i=1}^{80} h_i^u \right).$$ (3)

## 5. Implementation Details

The model was trained for 50,000 steps with a batch size of 4, using the AdamW optimizer and a cosine scheduler with a warm-up phase of 20,000 steps and a maximum learning rate of $10^{-4}$. Training was conducted on a GPU with 48GB of memory.

## 6. Experimental Results

### 6.1. Quantitative results

| Dataset | Method | AUROC | Accuracy | F1 Score | W. F1 Score | Precision |
|---|---|---|---|---|---|---|
| CT-RATE | Random Predictions | $49.88 \pm 0.62$ | $49.89 \pm 0.31$ | $27.78 \pm 0.51$ | $33.13 \pm 0.33$ | $49.85 \pm 1.12$ |
| | **3D CNN** | $76.49 \pm 0.28$ | $73.22 \pm 0.50$ | $46.86 \pm 0.31$ | $51.70 \pm 0.27$ | $38.46 \pm 0.54$ |
| | **CT-ViT** | $73.92 \pm 1.17$ | $70.83 \pm 0.17$ | $45.01 \pm 0.85$ | $49.65 \pm 0.88$ | $35.59 \pm 0.46$ |
| | **Swin3D** | $\underline{79.94} \pm 0.15$ | $75.95 \pm 0.25$ | $50.64 \pm 0.25$ | $54.68 \pm 0.21$ | $42.07 \pm 0.56$ |
| | **CT-Net** | $79.37 \pm 0.27$ | $\underline{77.37} \pm 0.40$ | $\underline{51.39} \pm 0.50$ | $\underline{56.37} \pm 0.32$ | $\underline{43.51} \pm 0.68$ |
| | **CT-Scroll** (ours) | $\mathbf{81.80} \pm 0.22$ | $\mathbf{79.49} \pm 0.45$ | $\mathbf{53.97} \pm 0.21$ | $\mathbf{58.08} \pm 0.28$ | $\mathbf{48.34} \pm 1.49$ |
| Rad-ChestCT | Random Predictions | $49.68 \pm 0.55$ | $50.40 \pm 0.32$ | $35.91 \pm 0.41$ | $47.72 \pm 0.51$ | $51.51 \pm 0.75$ |
| | **3D CNN** | $64.22 \pm 0.37$ | $57.08 \pm 0.90$ | $44.55 \pm 0.20$ | $56.21 \pm 0.34$ | $43.38 \pm 0.41$ |
| | **CT-ViT** | $63.31 \pm 0.98$ | $60.39 \pm 1.13$ | $45.13 \pm 1.42$ | $57.75 \pm 0.40$ | $41.48 \pm 0.40$ |
| | **Swin3D** | $67.29 \pm 0.23$ | $\underline{60.67} \pm 0.60$ | $\underline{47.98} \pm 0.41$ | $59.80 \pm 0.54$ | $\underline{44.03} \pm 0.45$ |
| | **CT-Net** | $\underline{67.71} \pm 0.83$ | $60.05 \pm 1.93$ | $47.53 \pm 0.93$ | $\mathbf{60.27} \pm 0.92$ | $43.38 \pm 1.19$ |
| | **CT-Scroll** (ours) | $\mathbf{71.21} \pm 0.37$ | $\mathbf{63.02} \pm 0.93$ | $\mathbf{48.55} \pm 0.54$ | $\underline{59.99} \pm 0.58$ | $\mathbf{48.35} \pm 0.49$ |

Table 1: Quantitative evaluation on the CT-RATE and Rad-ChestCT test sets. Reported mean and standard deviation metrics were computed over 5 independant runs. **Best** results are in bold, second best are underlined.

We evaluate the model's performance using standard metrics: AUROC, F1-Score, precision, and accuracy. For classification, we determine the threshold that maximizes the F1-Score for each of the 18 labels on the validation set, as F1-Score is the harmonic mean of precision and recall (Rainio et al., 2024). On the test set, we compute the average of each metric across all labels, as well as the weighted average F1 Score (W. F1 Score) based on label frequencies in the test set. Reported mean and standard deviation metrics were computed over five independent runs with different random seeds to ensure robustness. As shown in Table 1, our method achieves an F1-Score of 53.97 ($+\Delta 5.02\%$ over CT-Net) and an AUROC of 81.80 ($+\Delta 3.06\%$ over CT-Net) on the CT-RATE test set. On the Rad-ChestCT test set, CT-Scroll achieves a $+\Delta 12.47\%$ improvement in AUROC over CT-ViT, a $+\Delta 3.34\%$ increase over Swin3D and a $+\Delta 3.24\%$ increase compared to CT-Net. A paired t-test between our method and CT-Net on all metrics yielded p-values below 0.01, demonstrating the statistical significance of these improvements.

## 6.2. Ablation study

**Impact of the Scrolling Block module.** To evaluate the effectiveness of the proposed Scrolling Block, we compare its performance against various traditional modules by replacing the Scrolling Block with these alternatives. Table 2 presents the performance of our models and the contribution of each architectural component. Replacing a small 3D convolutional layer (Draelos et al., 2021) with a Global Average Pooling layer (Li et al., 2023) for dimensionality reduction yields a $+\Delta 2.39\%$ improvement in AUROC. Incorporating self-attention via Transformer Encoders enables long-range interactions between visual tokens from triplet slices, improving the F1-Score to 53.32, marking a $\Delta+1.25\%$ increase over the baseline without self-attention. Integrating local attention via a standard Sliding Window Attention mechanism (Beltagy et al., 2020), after an initial global attention module, leads to a $\Delta+0.64\%$ improvement in AUROC and a $\Delta+0.62\%$ increase in F1-score compared to the global-attention-only configuration. Introducing local attention limits the interaction between CT scan slices within the same spatial neighborhood, which could enable the model to learn more fine-grained feature representations, ultimately enhancing anomaly classification performance. Finally, incorporating the Scrolling Block leads to an AUROC of 81.80 ($+\Delta 0.68\%$ increase over global-attention-only configuration) and an F1-score of 53.97 ($+\Delta 1.22\%$ increase over global-attention-only configuration). Inference takes approximately 90 milliseconds, making it suitable for clinical practice applicability.

Table 2: Comparison of performance across different modules. We use Transformer Encoders with matching layer counts and computational costs to ensure fair comparisons across setups. *# Params* corresponds to the number of trainable parameters (in millions). *GPU Mem.* refers to the GPU Memory requirement per sample (in GB). *FLOPs* refers to the number of floating-point operations (in tera).

| Method | Reduction | Interactions | AUROC | F1 Score | # Params | GPU Mem. | FLOPs |
|---|---|---|---|---|---|---|---|
| 3D CNN | - | - | $76.49 \pm 0.28$ | $46.86 \pm 0.31$ | 0.3 | 26 | 0.388 |
| CT-ViT | - | - | $73.92 \pm 1.17$ | $45.01 \pm 0.85$ | 37 | 8 | 0.500 |
| Swin3D | - | - | $79.94 \pm 0.15$ | $50.64 \pm 0.25$ | 28 | 14 | 0.905 |
| CT-Net | **3D Conv.** | None | $79.37 \pm 0.27$ | $51.39 \pm 0.50$ | 15 | 15 | 1.344 |
| - | **Linear Proj.** | None | $80.42 \pm 0.51$ | $52.43 \pm 0.71$ | 70 | 15 | 1.345 |
| - | **GAP** | None | $81.21 \pm 0.40$ | $52.66 \pm 0.41$ | 12 | 15 | 1.335 |
| - | **GAP** | **Tr. Enc. (causal att.)** | $81.45 \pm 0.21$ | $52.98 \pm 0.44$ | 16 | 15 | 1.337 |
| - | **GAP** | **Tr. Enc. (global att.)** | $81.25 \pm 0.07$ | $53.32 \pm 0.22$ | 16 | 15 | 1.337 |
| - | **GAP** | **Tr. Enc. (global + local)** | $\underline{81.77} \pm 0.06$ | $\underline{53.65} \pm 0.39$ | 16 | 15 | 1.337 |
| CT-Scroll | **GAP** | **Scrolling Block** | $\mathbf{81.80} \pm 0.22$ | $\mathbf{53.97} \pm 0.21$ | 16 | 15 | 1.337 |

**Impact of the Sliding Window Size.** Table 3 presents the model's performance across various window sizes. CT-Scroll with a window size of 16 outperforms the wider window size of 64 by $\Delta+1.12\%$ in F1-score.

**Qualitative results.** Figure 4 illustrates CT axial slices with Grad-CAM activation maps (Selvaraju et al., 2019), extracted from the ResNet of the triplet slices embedding module, highlighting CT-Scroll's ability to identify abnormalities from relevant regions.

Table 3: Impact of the sliding window size. The sliding window size corresponds to the number of triplet slices considered in the attention computation.

| Window size | AUROC | Accuracy | F1 Score | Precision | Recall |
|---|---|---|---|---|---|
| **4** | $\underline{81.54} \pm 0.09$ | $78.94 \pm 0.37$ | $\underline{53.47} \pm 0.03$ | $46.99 \pm 0.48$ | $\mathbf{65.79} \pm 0.97$ |
| **16** | $\mathbf{81.80} \pm 0.22$ | $\mathbf{79.49} \pm 0.85$ | $\mathbf{53.97} \pm 0.21$ | $\mathbf{48.34} \pm 1.49$ | $\underline{65.36} \pm 1.21$ |
| **64** | $81.46 \pm 0.14$ | $\underline{79.42} \pm 0.72$ | $53.37 \pm 0.27$ | $\underline{47.75} \pm 0.84$ | $64.19 \pm 1.44$ |

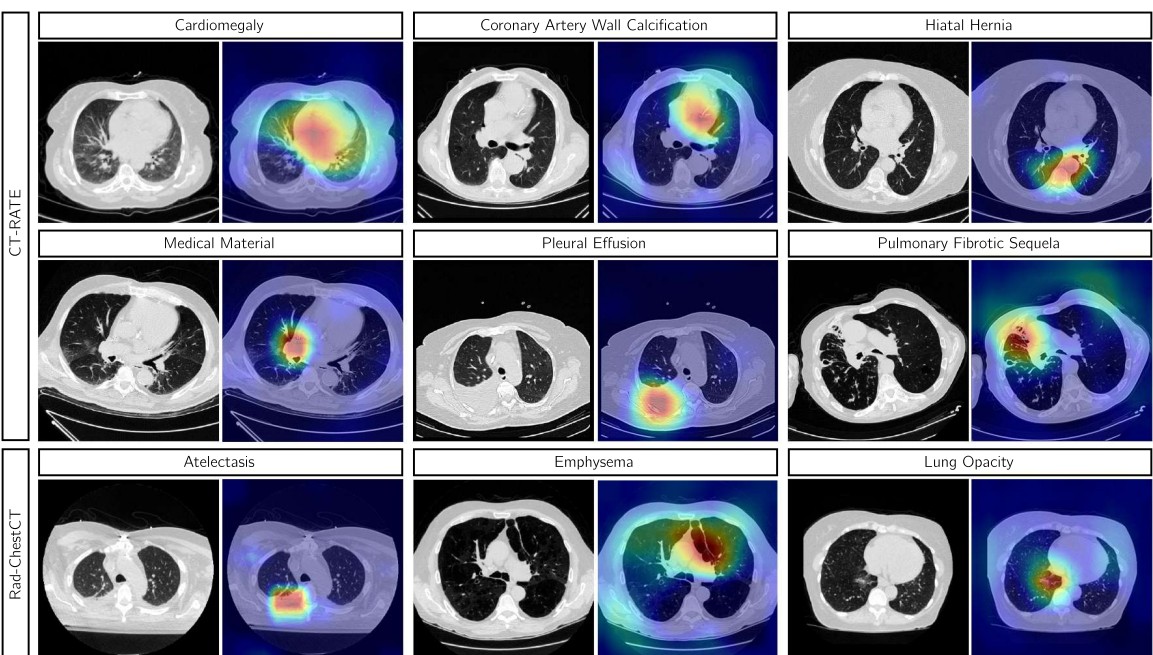

Figure 4: Grad-CAM activation maps from the last convolutional layer of the ResNet backbone within the Triplet Slices Embeddings block.

## 7. Conclusion

In this work, we introduce CT-Scroll, a hybrid model for multi-label classification from 3D CT Volumes. Specifically, our approach extracts triplet slices via a 2D CNN to capture fine-grained features, followed by an alternating global-local attention module that models both short- and long-range dependencies along the z-axis. CT-Scroll is evaluated on two public datasets, demonstrating improved multi-label classification performance while maintaining computational efficiency. Additionally, we perform an ablation study analyzing different modules for feature reduction and aggregation. Future work could explore the integration of region-specific information to further enhance classification performance, including the investigation of learnable fusion weights for features extracted from different window sizes. Moreover, taking full advantage of the third dimension by incorporating a 3D CNN module could help to preserve spatial continuity in volumetric data.

## Acknowledgments

We acknowledge (Hamamci et al., 2024a) for providing the CT-RATE dataset and (Draelos et al., 2021) for providing the Rad-ChestCT dataset. This work was performed using HPC resources from GENCI–[IDRIS] [Grant No. 103718]. We thank the support team of Jean Zay for their assistance.

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
