# OpenReview forum: "Imitating Radiological Scrolling: A Glocal-Lobal Attention Model for 3D Chest CT Volumes Multi-Label Anomaly Classification"
_MIDL.io/2025/Conference — MIDL 2025 Oral_

### Official Review · Reviewer_Te9Q · 2025-02-17

**Confidence:** 4
**Preliminary Rating:** 4
**Recommendation:** Oral
**Final Rating:** 5

**Summary:**

This paper emulates the scrolling behavior of radiologists by designing scrolling blocks to perform attention and later aggregates the multi-scale features to capture local abnormalities. Experiment results seem to be good.

**Strengths:**

1. The problem is worth investigating and leads to wide potential applications.
2. Emulating natural scrolling behavior using multi-scale attention is a very reasonable idea.
3. The results seem to be good.

**Weaknesses:**

Currently, I maintain my rating of 4 for the following reasons:
1. Cross-dataset generalization. I am curious about the generalization performance of the proposed model. If the proposed model pre-trained on CT-RATE can have reasonable performance on Rad-ChestCT, it will further demonstrate the model’s generalization ability: it is not learning the bias originated in the dataset.
2. Ablation studies on feature fusion weights of different windows. I would like to see if assigning different weights (e.g., in the feature fusion, assigning weights for different windows to be learnable) has the potential to further increase the model performance (since radiologists are 'examining the local content to confirm the diagnosis,' which I think the determination of disease can originate from information in one window).
3. Ablations on different image encoders: Since the emulation of the scrolling behavior largely relies on the multi-window fusion, this leads me to wonder if the model encoder can outperform previous multi-scale learning architecture, e.g., UNet.

**Detailed Comments:**

I am not sure whether there is a typo in the title: glocal-lobal → global-local. Please correct me if I am wrong.

**Justification Of The Final Rating:**

This paper provides a novel multi-window feature aggregation method that emulates human natural viewing behavior, and the problem investigated can lead to many meaningful applications. What makes me raise my score is the added Grad-CAM activation maps that demonstrate the interpretability of the method and the discussion with Reviewer 4PYZ. Overall, I recommend this paper to MIDL for its good performance and the potential to motivate future deep learning method design by emulating natural human behavior.

**Justification Of The Preliminary Rating:**

This paper investigates a problem that could lead to wide applications. The idea of integrating transformer blocks to emulate human behavior could lead to new designs that can boost model performance. However, utilizing multi-scale information for medical imaging is not a completely novel idea (e.g., UNet, fMRI foundation models that use multi-scale attention and masking); it would be fascinating to see if their multi-scale attention places more importance on key ROIs that make radiologists confirm diagnoses, emulating human behavior.

**Questions To Address In The Rebuttal:**

Please refer to the weakness part.

**Special Issue:**

Yes

---

> ### Author Response · Authors · 2025-03-07
>
> We would like to sincerely thank the reviewer for the time spent examining our work and for providing valuable comments and suggestions. These insights have helped us improve the clarity of our manuscript and have offered clear directions for future research.
>
> Question 1.
> For all experiments, models are trained on the CT-RATE train set.  We extend our experimental results with an evaluation of these models on the Rad-ChestCT test set (The train set from Rad-ChestCT dataset is not available). To improve clarity, we have revised Section 3 to explicitly highlight our cross-dataset evaluation strategy.
>
> Question 2.
> We thank the reviewer for this suggestion. The learnability of feature fusion weights for different windows is indeed a promising direction, as it could allow the model to extract features at multiple scales, potentially benefiting the detection of specific anomalies. A particularly interesting extension would be to incorporate a Mixture of Experts module, enabling the model to learn which window should be prioritized for different cases in a data-driven manner, aligning with the idea that certain windows may carry more diagnostic value for specific anomalies.
> However, due to the time constraints of the rebuttal period and our available resources, we are unable to conduct these additional experiments at this stage. We greatly appreciate this perspective and will certainly consider it for future work. Additionally, we have updated our conclusion to reflect this consideration.
>
> Question 3.
> We thank the reviewer for this insightful suggestion. If we correctly interpret the comment, the proposition suggests to evaluate the performance of the CT-Scroll encoder on downstream tasks such as detection or segmentation and compare it with established multi-scale architectures like U-Net. We appreciate this valuable direction and agree that it could provide further insights into the model's generalization ability. However, due to time constraints during the rebuttal period, we were unable to conduct these additional experiments. We plan to explore this comparison in future work.
>
> Detailed Comments: We thank the reviewer for the remark, we corrected the title with "Global-Local".
>
> We are grateful for the thorough feedback provided and we thank again the reviewer for their valuable remarks.

---

> > ### Comment · Reviewer_Te9Q · 2025-03-11
> >
> > Dear authors, I have read your response, and it addressed part of my concerns. From the novelty of utilizing multiple windows to emulate human behavior and the good experiment results, I recommend this paper to MIDL in my finalized score.

---

### Official Review · Reviewer_4PYZ · 2025-02-18

**Confidence:** 4
**Preliminary Rating:** 3
**Recommendation:** Poster
**Final Rating:** 4

**Summary:**

This work introduces a hybrid model for 3D CT volumes that extracts triplet slice embeddings using a 2D CNN and facilitates interactions between these representations through both global and local attention mechanisms, mimicking the behavior of a radiologist. The approach is evaluated on two public datasets: CT-RATE and Rad-ChestCT.

**Strengths:**

1. The writing and structure of this paper are straightforward.
2. The paper includes numerous tables and figures.
3. The paper employs multiple datasets for experiments and analysis.
4. The model design presented by the authors is intuitive.
5. The authors tell a good story in the paper.

**Weaknesses:**

1. The authors reviewed a significant number of previous studies in the related work section but did not provide sufficient explanation or comparison regarding the differences and relationships between the method proposed in this paper and the existing works. Do these previous methods have any limitations? If so, are these limitations addressed in the current work?
2. Is the dataset preprocessed consistently for all the methods being compared? Specifically, are the baseline models also subject to the same preprocessing steps to ensure a fair comparison?
3. The authors mention using Global Average Pooling (GAP) instead of a linear projection or a 3D convolutional layer for dimensionality reduction. However, there seems to be no ablation study for the linear projection component. Why was this part of the design not tested in an ablation experiment?
4. Instead of relying solely on ResNet-18 as the backbone, would the authors consider experimenting with a wider variety of backbones to further validate the proposed method's effectiveness across different architectures?
5. Regarding the comparison with the baseline (GAP), the improvements in the ablation studies are relatively small in terms of performance, while the increase in parameters (M) and inference time (ms) is much larger. Does this suggest a trade-off compared to the baseline? How do the authors justify this trade-off?
6. Regarding the loss function, did the authors perform ablation tests with alternative loss functions, or experiment with hybrid loss functions? What were the considerations behind selecting the current loss function?
7. In the comparison studies (Quantitative evaluation on the CT-RATE and Rad-CT-Chest), the authors should provide more comprehensive metrics such as parameters (M), FLOPs (T), inference time (ms), and training time (as stated in the contributions) to offer a more balanced comparison with the baseline models.
8. The paper does not adequately explain or justify why the proposed method achieves state-of-the-art (SOTA) performance while maintaining a lightweight structure. Could the authors provide more details on how these factors are balanced?
9. While the proposed method tells a good story, it appears to be a straightforward integration of existing technologies. Combining current modules in this manner does not seem to be a novel approach, as this design pattern is already widely used. Could the authors further clarify what sets their approach apart from prior work?
10. The authors should include a comparison of Grad-CAM results between the baseline models and the proposed method to provide further insight into the model’s interpretability and feature importance.
11. How much spatial information is lost when 3D voxels are processed into 2D slices? What do the authors think about the 3D information that is not captured by these 2D slices?
12. The summary of the core contributions of this paper is relatively weak and lacks sufficient detail.

**Detailed Comments:**

Please refer to the weaknesses section.

**Justification Of The Final Rating:**

Dear authors, I have read your response to the reviewers. Compared to the earlier version, this revision includes clearer instructions and additional clarifications. The workload and explanation now appear more reasonable, and I can recommend this paper to MIDL.

**Justification Of The Preliminary Rating:**

While the authors have made efforts to propose a novel method, key aspects of the work—such as methodological comparison, experimental rigor, and the justification of design choices—need further clarification and discussion.

**Questions To Address In The Rebuttal:**

Please refer to the weaknesses section.

**Special Issue:**

No

---

> ### Author Response · Authors · 2025-03-07
>
> We would like to thank the reviewer for the time spent reviewing our paper, as well as for the insightful comments and suggestions that have greatly contributed to improving the clarity of our work and have also provided valuable directions for future research.
>
> Question 1.
> We acknowledge that while our Related Work section provides a broad overview of previous studies, a more explicit discussion of their limitations in relation to our approach is indeed beneficial.
> To address this, we have expanded Section 2.1 (Related Work) to clearly articulate the constraints of prior methods and how our approach mitigates them. Specifically, convolutional networks are inherently limited by their local receptive fields, restricting their ability to capture global contextual dependencies. Vision Transformers alleviate this by leveraging attention mechanisms that enable interactions across different regions of the image, thus capturing long-range dependencies. Swin Transformers, also attention-based, introduce a windowing mechanism to focus on local neighbourhoods, improving their ability to model shorter-range dependencies. However, these fully attention-based architectures are computationally expensive and generally require extensive pretraining on large-scale datasets.
> CT-Net addresses this challenge by employing 2D slices processed through a ResNet, followed by a 3D convolutional network. While this approach demonstrates strong performance, it remains fully convolutional and thus inherits the limitations of 3D CNNs, particularly in capturing broader contextual information. Our proposed CT-Sroll mitigates these issues by first leveraging a ResNet to extract fine-grained details from triplet slices using convolutional layers. To further enhance contextual understanding, we introduce a global attention module, allowing triplet slices to interact and capture broader anatomical structures. Additionally, our local attention mechanism, implemented via Sliding Window Attention, effectively manages short-range dependencies along the z-axis, ultimately enhancing multi-label classification performance.
> We have revised the Related Work section to explicitly detail these limitations and highlight how our method overcomes them. We appreciate the reviewer’s insightful suggestion and believe this refinement strengthens the clarity and positioning of our contribution.
>
> Question 2.
> We confirm that both the CT-RATE and Rad-ChestCT datasets are preprocessed consistently across all methods to ensure a fair comparison. To further clarify this, we have revised the "Preprocessing" section in Section 3 to explicitly detail the preprocessing steps applied uniformly to all models.
>
> Question 3.
> We appreciate the reviewer’s insightful comment. In response, we have conducted an additional experiment incorporating a Linear Projection for dimensionality reduction applied to each feature map, and the results have been added to Table 2. Our findings indicate that while Linear Projection outperforms the 3D CNN alternative, it remains inferior to Global Average Pooling (GAP) in terms of multi-label classification performance. Furthermore, GAP achieves this without introducing additional parameters, reinforcing our design choice.
>
> QuestIon 4.
> Our primary focus in this work is on aggregation rather than feature extraction. We selected ResNet-18 as a backbone due to its well-established balance between complexity and performance, ensuring a fair and efficient evaluation of our method within the constraints of paper length and computational resources. That said, we acknowledge the value of exploring a broader range of backbones to further validate our approach. As part of future work, we plan to investigate alternative architectures and pretraining strategies, including leveraging 2D visual encoders pretrained on different modalities, such as X-ray images, to enhance performance beyond natural image representations.
>
> Questions 5 & 8.
> While the current improvements in performance over the baseline may appear modest, it is crucial to note that existing methods do not yet achieve clinically sufficient performance levels. In the development of our proposed method, we consider that it is essential to incrementally enhance performance to approach optimal levels. Once these optimal performance thresholds are attained, we will then focus on striking a balance between performance and computational complexity. This iterative process ensures that we first address the critical need for improved accuracy before optimizing for efficiency.
>
> Question 6.
> To ensure a fair comparison across configurations, we employed a standard binary cross-entropy loss function throughout all experiments, as our primary focus is on feature reduction and aggregation. Exploring alternative or hybrid loss functions is an interesting direction for future work and may further enhance performance. We appreciate the suggestion and will consider it in future studies.

---

> > ### Author Response · Authors · 2025-03-07
> >
> > Question 7.
> > These metrics (Parameters count, FLOPs and inference time) associated with baseline models are provided in the first rows of Table 2. Additionally, we have included a “GPU Mem.” column to report the GPU memory consumption per sample.
> >
> > Question 9.
> > Our hybrid approach leverages both convolutional neural networks and transformer-based attention, and its novelty lies in how we integrate these components to enhance multi-label anomaly classification from 3D CT volumes. Existing methods that rely solely on convolutional networks or attention mechanisms each have inherent limitations, as discussed in our responses to Questions 1. Our approach is inspired by CT-Net, utilizing triplet axial slices and ResNet-based feature extraction to capture fine-grained details via a 2D CNN. Instead of employing a computationally expensive 3D network, we introduce a Global Average Pooling layer to reduce the dimensionality of extracted features, improving classification performance without adding additional parameters (see Table 2, Li et al. 2023, Section 4.1). Furthermore, our feature aggregation strategy draws inspiration from recent advancements in Natural Language Processing, where alternating global and local attention has proven effective in capturing both short- and long-range dependencies. We adapt this concept to the medical imaging domain by treating a 3D CT volume as a sequence of visual tokens (triplet axial slices along the z-axis). Our ablation study demonstrates the effectiveness of this global-local attention mechanism, highlighting its ability to better model contextual dependencies and enhance multi-label classification performance in this domain.
> >
> > We appreciate the reviewer’s feedback, which has helped improve clarity. To address this, we have added clarifying sentences in the Abstract and Related Work (Section 2.1).
> >
> > Question 10.
> > We appreciate the reviewer’s insightful suggestion. In response, we have included Figure 4, which presents Grad-CAM activation maps highlighting relevant regions for anomaly detection across volumes from both the CT-RATE and Rad-ChestCT test datasets. This visualization demonstrates the ability of our proposed method to focus on meaningful features. While we acknowledge the value of comparing these activation maps with baseline models, time constraints prevented us from conducting this analysis. We consider this an important direction for future work.
> >
> > Question 11.
> > Indeed, there is a potential loss of spatial continuity when processing 3D voxels as 2D Slices instead of using a full 3D CNNs. However, our results, which compares our method with a 3D CNN, suggest that there is enough spatial information contained in the way we process the 3D volume with 2D slices to perform well in the considered multi-label classification problem.
> > While our approach leverages 2D slices to balance computational efficiency and performance, an interesting direction for future work would be to develop a hybrid model that combines both a CNN branch for feature extraction from the 3D Volume and another one that would extract feature from a 2D slice-based processing. By aggregating features from these complementary branches, such an approach could mitigate the loss of 3D structural information while preserving the advantages of both representations.
> > We appreciate the reviewer’s insightful question, which has helped refine our discussion of this potential research direction in our Conclusion (Section 7).
> >
> > Question 12.
> > We have revised the summary of our core contributions in Section 1 to provide greater clarity and detail. Specifically, we now emphasize the feature reduction and aggregation modules, as well as the impact of the spatial extent of the attention mechanisms. We appreciate the reviewer’s feedback, which has helped us improve the structure and clarity of this section.
> >
> > We again sincerely thank the reviewer for their valuable suggestions, which have significantly enhanced the quality of our work.

---

> ### Comment · Area_Chair_uLuP · 2025-03-14
>
> Dear Reviewer,
>
> The discussion period ends in less than 24 hours, and your final rating is missing. Please review the authors’ response, revisions, and peer feedback, then update your score. You can submit your final rating along with the justification by editing your original review.
>
> Your final rating is critical for the decision.
>
> Thanks,
> AC, MIDL 2025

---

### Official Review · Reviewer_NcAV · 2025-02-28

**Confidence:** 2
**Preliminary Rating:** 4
**Recommendation:** Poster
**Final Rating:** 4

**Summary:**

This paper proposes CT-Scroll, an innovative deep learning model aimed at multi-label anomaly classification in 3D chest CT scans. The key idea is to mimic the radiologist’s behavior when reviewing CT volumes.

**Strengths:**

The idea of emulating the radiologist’s scrolling behavior is both novel and intuitively appealing. The integration of global and local attention mirrors the diagnostic process in clinical practice.
The paper presents thorough experiments on two datasets with clear improvements in key performance metrics. The ablation studies provide solid evidence on the effectiveness of the model components.

**Weaknesses:**

In section 4.2. Scrolling Block. need to be explained more, such as with detailed diagrams or pseudocode.

The term “glocal-local” and the underlying intuition behind alternating global and local attention layers need clearer articulation. While the concept of mimicking radiologist scrolling is appealing, the paper could more rigorously explain why and how this specific attention alternation improves feature representation compared to a unified attention strategy.

**Detailed Comments:**

Add more detailed explanation to the scrolling block since it is the key innovation in this paper

**Justification Of The Final Rating:**

Thanks for your response. The response has already addressed my concern. Simulating radiologists by using scrolling blocks enhances the understanding of CT images. I would like this paper to be accepted by MIDL.

**Justification Of The Preliminary Rating:**

This paper makes a great contribution by introducing a model that effectively combines global and local attention to mimic the scanning behavior of radiologists. The experimental results are strong, showing significant improvements over baselines, while ablation studies provide valuable insights into the model's architecture.

**Questions To Address In The Rebuttal:**

Can you explain more about the "Scrolling Block"?

---

> ### Author Response · Authors · 2025-03-07
>
> We sincerely appreciate the reviewer’s time and thoughtful feedback, which has helped us refine the clarity and presentation of our work.
>
> We have revised Section 4.2 to provide a clearer explanation of the Scrolling Block, including additional references to attention masks in Figure 3 and improved annotations in Figure 2. Below, we detail the underlying motivation and implementation of our approach.
>
> A 3D CT scan is processed as triplets of axial slices, each encoded into a feature representation using a 2D convolutional network to extract fine-grained localized features (Section 4.1).
>
> Radiologists typically scroll through axial slices to build a global understanding of the scan by observing anatomical structures and contextualizing abnormalities. To model this behavior, we introduce a Transformer Encoder with global attention as the first component of the Scrolling Block. This allows feature representations of triplet slices to interact across the entire volume, capturing long-range dependencies along the z-axis (Section 4.2).
>
> Beyond this global context, radiologists refine their focus by scrolling back and forth around regions of interest, reinforcing their diagnostic decisions through localized context exploration. We model this behavior with two additional Transformer Encoders with local attention:
> - The first local attention block captures bottom-up (caudal-to-cranial) interactions, where each triplet slice attends to those above it (Figure 3.b).
> - The second local attention block captures top-down (cranial-to-caudal) interactions, attending to triplet slices below (Figure 3.c).
>
> These local attention mechanisms enable the network to capture short-range dependencies along the z-axis, refining the feature representation in a way that aligns with human diagnostic patterns. The corresponding attention masks are illustrated in Figure 3 (a) and (b).
>
> To improve clarity, we have made the following updates:
> - Enhanced Section 4.2 with explicit explanations of short- and long-range dependencies.
> - Updated Figure 3 to reference specific attention masks used in each block.
> - Refined Figure 2 by adding function notations (f_SB for the scrolling block, f_G for the Global Attention, f_CAU->CRA and f_CRA->CAU for the local attention blocks).
> - We also expanded the Relation Work (Section 2.1) to better highlight the limitations of existing methods and the advantages of alternating global and local attention for multi-label classification from 3D CT Volumes.
>
> These revisions clarify the intuitive and technical motivations behind our glocal-local attention strategy and its role in improving feature representation.
>
> We thank the reviewer once again for their insightful feedback, which has helped us improve the clarity of our work.

---

### Author Rebuttal · Authors · 2025-03-07

**Rebuttal:**

We sincerely thank the reviewers for their valuable time and thoughtful feedback, which has significantly improved the quality of our submission. All modifications in response to the reviewers' comments are highlighted in red, in the supporting material pdf.

**Supporting Material:**

/attachment/f13c45e9cefe6c3331b60080ccea4b7ecd6688ea.pdf

---

### Meta-Review · Area_Chair_uLuP · 2025-03-23

**Recommendation:** Accept (Oral)
**Confidence:** 5

**Metareview:**

This paper received one strong accept and two weak accepts. All reviewers acknowledged its practical relevance, intuitive design, and strong empirical results. The authors provided detailed and thoughtful responses that addressed key concerns, including architectural clarity, generalization, and interpretability. Based on the positive consensus and substantive revisions, I recommend acceptance.